# After-Effects of Hydrochar Amendment on Water Spinach Production, N Leaching, and N_2_O Emission from a Vegetable Soil under Varying N-Inputs

**DOI:** 10.3390/plants11243444

**Published:** 2022-12-09

**Authors:** Haijun Sun, Ying Chen, Zhenghua Yi

**Affiliations:** Co-Innovation Center for Sustainable Forestry in Southern China, College of Forestry, Nanjing Forestry University, Nanjing 210037, China

**Keywords:** biochar, leachate, N losses, nitrous oxide, vegetable production

## Abstract

Biochar use in agriculture brings significant agronomic and environmental co-benefits, which are a function of biochar and crop types and nitrogen (N) rates. We here conducted a soil column experiment to evaluate the after-effects of hydrochar amendment at 0.5 and 2.0 wt% on vegetable production, N recovery and losses via leaching and nitrous oxide (N_2_O) emission from water-spinach (*Ipomoea aquatica* Forsk)-planted vegetable soil receiving three N inputs (120, 160, and 200 kg/ha). The results showed that hydrochar with 2.0 wt% significantly (*p* < 0.05) improved the biomass yield of water spinach, receiving 120–160 kg N/ha by 11.6–14.2%, compared with no change in the hydrochar treatment. Hydrochar had no effect on total N content of water spinach, and only increased the total N recovery under 2.0 wt% given hydrochar amended treatment with 120 kg N/ha. Neither pH or EC of leachate was changed with N reduction or hydrochar application. However, in some cases, hydrochar changes the NH_4_^+^, NO_3_^−^ and total N concentrations in leachate. When applied at 2.0 wt%, hydrochar significantly (*p* < 0.05) increased total N leaching losses by 28.9% and 57.1%, under 120 and 160 kg N/ha plot, respectively. Hydrochar applied at two rates increased the N_2_O emissions by 109–133% under 200 kg N/ha but decreased them by 46–67% under 160 kg N/ha. Therefore, after three years of application, hydrochar still improves the production of leafy vegetable, but the impacts on N leaching and N_2_O emission vary, depending on inorganic N and hydrochar application rates.

## 1. Introduction

In recent years, the demand for vegetables has increased because of changes in dietary habits worldwide. In particular, the total vegetable plantation area in China has significantly increased over the past three decades. Meanwhile, nitrogen (N) fertilization is a widespread practice in the management of vegetable fields to ensure biomass yield and quality of marketable produce [1]. However, farmers have always applied excessive N inputs to vegetable production, which led to low agronomic N use efficiency [2,3] (Bai et al., 2020; Qasim et al., 2020). According to some previous investigations, the vegetable N use efficiency is typically below 30% [4], particularly in intensive vegetable systems, where N use efficiencies of only 12–18% are the norm [5,6]. Consequently, the gains in vegetable production have been associated with substantial agricultural N losses, which contribute to water pollution, greenhouse gas emissions and damage to human health [7], of which, leaching and nitrification-denitrification are two main N loss pathways in vegetable production. In some intensive vegetable production regions in China (e.g., Shouguang, Shandong province), nitrate (NO_3_^−^-N) leaching losses of 350–550 kg/ha have been recorded [8,9]. Globally, the emission of N_2_O from soils of open-field vegetable production systems has been estimated to be 0.95 Tg per year [10]. Therefore, maintaining vegetable production while reducing the detrimental effects of N application is an urgent priority for global food security and environmental sustainability.

Biochar (BC), a carbon production from biomass pyrolysis, is beneficial for improving soil quality and crop productivity [11] and reducing N losses [12,13]. Biochar has been proven to mitigate N_2_O emissions along with the increase of denitrifiers in vegetable soil [5] (Li et al., 2017). Li et al. (2020) found that 20 t/ha can be an optimal biochar rate in mitigating yield-scaled N_2_O emissions from Ultisols planted with vegetables [14]. Although biochar may generally reduce N_2_O emissions and N leaching, inconsistent results in response to biochar amendment have been observed [12,15], likely because of the difference in soil and biochar type as well as the cropping systems.

Biochars are mainly derived with two entirely different processes, i.e., pyrolysis and hydrothermal carbonization processes, which produce two biochars named pyrochar and hydrochar, respectively. And it has been observed that these two biochars have different properties [16,17]. The biochar properties such as ash content and liable organic compounds may induce transient effects that alter N-based losses. However, compared with pyrochar, given the effects of the addition of hydrochar into vegetable soils on the plant biomass, N losses are lacking [18,19]. In addition, biochar’s performance changes as the biochar progressively ages in soil, through chemical reactions, water erosion, and microbial decomposition [20,21]. For instance, a previous work concluded that the time-dependent impact on N_2_O emissions and NO_3_^−^ leaching is a crucial factor that needs to be considered in order to develop and test resilient and sustainable biochar-based N loss mitigation strategies [15] (Borchard et al., 2019). Water spinach (*Ipomoea aquatica* Forsk) is a commonly planted vegetable in Southeast Asia, which can be planted under both upland and flooding cultivation conditions [22,23]. To date, nevertheless, a limited number of studies have investigated the after-effects of hydrochar application into water-spinach-planted vegetable soil.

Therefore, the specific objectives of the current study were: (1) to evaluate the responses of production and N recovery of water spinach to hydrochar after being applied three years, and (2) to determine the reactive N losses via leaching and N_2_O emission as result of hydrochar applied at two rates. The results could provide more evidence to support the use of biochar in vegetable soil considering its agronomic and environmental effects.

## 2. Results

### 2.1. Production of Water Spinach

The water spinach shoot biomasses were harvested four times. At the first and second harvests, hydrochar only exerted positive effects under N120 when being applied with 2.0 wt%, i.e., the N120-BC-2.0% with 69% and 43%, significantly (*p* < 0.05) higher shoot biomasses than N120-BC0, respectively. With same N application rate, hydrochar had no significant effect on the water spinach production harvested at the third and fourth times. At the end of the experiment, when the inorganic fertilizer N reduced by 20–40% to 120–160 kg/ha, hydrochar applied with 2.0 wt% could significantly (*p* < 0.05) promote the productions of water spinach by 12–14% (Table 1).

### 2.2. The N Contents and Uptake Capacity of Water Spinach

At different harvest times, the water spinach had remarkably varied N contents. Overall, the highest N contents of water spinach for all treatments were recorded at the first harvest (Figure 1). However, the hydrochar application, whether being applied at rate of 0.5 or 2.0 wt%, did not change the total N contents of the water spinach that receiving equal N fertilizer that ranging from 120 to 200 kg/ha.

Under 200 and 160 kg N/ha input, both hydrochar amendments with two rates had no influence on the split and total N uptake capacities of water spinach that harvested at all four times (Table 2). However, the total N capacity was 23%, significantly (*p* < 0.05) higher under N120-BC-2.0% relative to N120-BC0 (279 and 227 mg/pot, respectively), which was mainly attributable to the significantly (*p* < 0.05) 73% increased biomass produced at the first harvest.

### 2.3. The Properties of Leachate

The leachate samples collected on 30 July 2021 were with relatively lower pH and EC values than those collected on 11 and 29 August 2021 (Figure 2). Under three N inputs, hydrochar amendments did not change either the pH or the EC of leachate, though there were some non-significant differences in EC.

From Figure 3, we found that the major forms were organic and NO_3_^−^-N was contained in the leachate. The NH_4_^+^-N were lower than them with only 0.28–0.40 mg/L, and varied slightly at three observations. The NO_3_^−^-N concentrations in leachate collected at 11 and 29 August 2021 were obviously higher than that collected on 30 July 2021 (1.44–2.44 and 1.68–2.67 mg/L vs. 0.22–0.56 mg/L, respectively). This trend was also found for the organic and total N concentrations. On 30 July and 11 August 2021, it seems that biochar addition had no significant effect on NO_3_^−^-N concentration, but increased the organic and total N concentrations in the leachate. However, on 29 August 2021, only under 120 kg N/ha, the organic and total N concentrations were increased following the hydrochar amendment.

The NH_4_^+^-N, NO_3_^−^-N, organic and total N leaching losses under three N inputs were 0.87–1.34, 3.42–6.52, 5.21–9.85 and 10.6–17.6 g/pot, respectively (Table 3). Under 200 and 160 kg N/ha, hydrochar addition at both rates had no influence on the NH_4_^+^-N and NO_3_^−^-N leaching losses. When inorganic N input further reduced to 120 kg N/ha, hydrochar addition still did not impact the NO_3_^−^-N, but significantly increased the NH_4_^+^-N by 37–46% (*p* < 0.05). Compared with BC0, BC-2.0% significantly increased the organic N leaching losses by 44%, 46%, and 89% under 200, 160, and 120 kg N/ha, respectively (*p* < 0.05). Hydrochar applied at 0.5 wt% only increased the NO_3_^−^-N leaching under 120 kg N/ha.

### 2.4. N_2_O Emission

We found that hydrochar addition did not change the N_2_O emission fluxes pattern in vegetable soil (Figure 4A). Under 200 kg N/ha, the biochar amendments with 0.5 wt% and 2.0 wt% significantly (*p* < 0.05) increased the total N_2_O emission by 108% and 132%, respectively (Figure 4B). Nevertheless, the total N_2_O emission under two biochar amendments were 45.5–67.0%, significantly lower than the no-biochar-added treatment, under N160 conditions (*p* < 0.05). Overall, there was no difference in total N_2_O emissions from the two hydrochar amended treatments under N200 and N160, respectively. Under N120, no difference was found between BC0 and BC-0.5%, but 48.5% higher N_2_O emission was found in BC-2.0%, compared with the BC0 treatment (Figure 4B).

## 3. Discussion

### 3.1. Hydrochar with High Rate Improves Production of Water Spinach under Reducing N Inputs

Increasing vegetable and/or biomass yield is often considered as the most important goal and the positive effect of biochar application [24,25]. Ren et al. (2022) summarized that the yields of wheat, maize and rice were shown to increase between 10% and 19% while N application rates were reduced by 15–19% [26]. Inconstantly, no change in water spinach biomass production was found when urea N reduced from 200 kg/ha to 120–160 kg/ha. Two of the main mechanisms for yield increase by biochar in upland crop may be a liming effect and an improved water holding capacity of the soil, along with improved crop nutrient availability [27]. Remarkably, the highest yield increases in spinach were achieved under high biochar (2.0 wt%) amended treatments with two reducing N inputs 120 and 160 kg/ha. Easily understood, 200 kg/ha was enough to meet the N demand of spinach to gain a high biomass production. Crop yields increase strongly if site-specific soil constraints and nutrient and water limitations are mitigated by appropriate biochar formulation [28]. Therefore, when the N nutrient is not the limitation on the plant growth, even biochar addition had no effect on further crop yield. However, no impact of biochar on biomass yield of water spinach was discerned when applied at low rate (0.5 wt%) in the current work (Table 1). Similar results can be found in Li et al. (2020), where 10 t/ha biochar addition had no effect in increasing vegetable yield [14]. Overall, the hydrochar effect on vegetable production is a function of its own and N input rates. In the current study, hydrochar at high rate (2.0 wt%) improves production of water spinach while reducing 20–40% the N inputs.

Biochar could increase the apparent recovery of N and therefore the vegetable yield [5,29] (Li et al., 2017; Murtaza et al., 2021). In the current work, under three N inputs, hydrochar at 0.5 or 2.0 wt% does not change the N content in water spinach that is harvested at four times (Figure 1). Consequently, with only 60%N input, BC-2.0% treatment shows significantly higher total N recovery than does the BC-0 treatment. This result indicates that enhanced N recovery partially explains the improved water spinach production in some treatments. Other underlying mechanisms about hydrochar’s effect on vegetable production, such as those reported in the literature [29,30], should be further confirmed in future investigations.

### 3.2. Hydrochar Potentially Increase the N Leaching Risk

Excessive application of chemical fertilizers has caused leaching of N from vegetable field into aquatic systems. One alternative method to reduce nutrient leaching is the application of biochar to soils. According to previous study, biochar can act as a carrier matrix for nutrients and therefore reduce N leaching losses [15]. Overall, soil NO_3_^−^ concentrations remained unaffected while NO_3_^−^ leaching was reduced by 13% with biochar according to a meta-analysis [15]. Nevertheless, our experiment found that hydrochar addition had no influence on NO_3_^−^ leaching loss. This is possibly a result of the difference in absorption capacity on NO_3_^−^ between varied char materials [31,32]. Moreover, at the lowest N input in current work, or being applied with higher rate, hyrochar increased N leaching from vegetable soil. This negative effect of hydrochar was mainly attributed to the higher organic N leaching losses (Table 3). Therefore, it is likely that hydrochar bound urea could reduce N leaching, as performed in a previous report [33]. What is more, modifying the hydrochar may be another approach to mitigation of N leaching in vegetable soil.

### 3.3. Effects of Hydrochar on N_2_O Emission Depend on N Rate

Although biochar may generally reduce N_2_O emissions, inconsistent results in response to biochar addition have been observed. For instance, biochar soil application reduces agricultural N_2_O emissions [11]. However, biochar applied below 10 t/ha had lower N_2_O mitigating effect, according to a meta-analysis [15]. These inconsistent responses were likely because of the variation in the physical and chemical characteristics of different biochar types, as well as differences in the studied soil types and cropping systems [34]. Moreover, different biochar amendment rates and modes of incorporation may contribute to different responses of N_2_O emissions from agricultural soil [14]. In the current work, hydrochar exerted contrasting effects on N_2_O emission from vegetable soils with 200 kg N/ha and 160 kg N/ha. Moreover, the influences were independent of hydrochar application rates. It has not been previously reported that effects of hydrochar on N_2_O emission from vegetable soils as well as the plant disease [35], depend on N rate, a finding which was proved in the current work.

## 4. Materials and Methods

### 4.1. Soil and Hydrochar Characterization, Soil Column Installation

The tested soil was taken from three soil profiles (0–20, 20–40 and 40–60 cm, respectively) at a typical vegetable production base from Yixing (31°29′ N, 119°60′ E) of Jiangsu Province, China. The tested soil was classified as Anthrosol. Collected soil was naturally dried under airy conditions and was manually ground to pass through a 20-mesh nylon sieve. The soil was layer-wise repacked into PVC column with inner diameter 35 cm and height 70 cm. In each soil column, the soil was repacked in approx. similar bulk density according to the same profile order as that collected from the rice field. In total, there were approx. 35 kg of soil used per column. The soil at 0–20 cm layer was with an initial pH (1:2.5 soil to water) of 5.83, an electrical conductivity (EC in 1:5 soils to water extract) of 0.32 mS/cm, a total N content of 1.35 g/kg, a soil organic matter content of 23.8 g/kg, and consisted of 8.5% sand, 73.6% silt, and 17.9% clay.

Dr. Yanfang Feng, working at Jiangsu Academy of Agricultural Sciences, helped us to produce hydrochar for the current experiment. The detailed preparation of hydrochar was as follows: wheat straw and water were uniformly mixed at the ratio of 1:10 (*w*/*v*) and kept in a sealed autoclave at 260 °C for 1 h. After that, the HBC was obtained after solid-liquid separation using a filter, and then oven-dried (at 70 °C) to constant weight. The basic properties of applied hydrochar can be found in Feng’s published paper [16]. We homogeneously mixed the hydrochar with the 0–20 topsoil according to the following rates 12.5 and 50 t/ha (approx. corresponding to 0.5 and 2.0 wt% of the weight of 0–20 cm layer topsoil), to represent low and high application rates, respectively, on 25 July 2018. Thereafter, no hydrochar was added to the tested soils in the PVC column. Therefore, the current work evaluated the after-effects of hydrochar incorporated into the vegetable soil system.

### 4.2. Experiment Design and Crop Management

Three N inputs, 120, 160 and 200 kg/ha, were considered. For each N level, three hydrochar amending rates, namely, 0 wt% (BC0), 0.5 wt% (BC-0.5), and 2.0 wt% (BC-2.0) of the top layer (0–20 cm) soil, were investigated. For each treatment, three replicates were set up. Therefore, there were in total 27 soil columns managed outdoors in a nearly natural environment. The N fertilizations were done by the addition of urea (N content, 46%), which were equally split applied in soils as the basal and top dressing on12 July and 14 August 2021, respectively. Approx. 35 g/pot and 140 g/pot dry hydrochar were added to each char-added column with the top layer (0–20 cm) soil. The application rates of phosphorus and potassium fertilizers for all treatments were 100 kg P_2_O_5_/ha and 100 kg K_2_O/ha, which were done as basal fertilizers by the addition of calcium superphosphate and potassium chloride, respectively.

We used Meifeng 1 as the tested water spinach (*Ipomoea aquatica* Forsk.). The seeds of water spinach were sown in the soil directly and then thinned up to a height of 5–10 cm after twelve days. There were six plants in each soil column, and they were grown under upland conditions until harvest.

### 4.3. Sampling and Measurements

#### 4.3.1. Plant Sample and Its Total N Content Determination

The plants of water spinach were sampled four times, on 30 July, 13 August, 1 September, and 20 September 2021. At each harvest, plants were carefully harvested from the soil column. After fresh weights were measured, they were dried at 80 °C for 48 h to record the dry weights. The N content of plant materials was determined by the semi-micro Kjeldahl method [36].

#### 4.3.2. N_2_O Emission

We used the static chamber-gas chromatography method to determine the N_2_O concentration of gas samples. Briefly, the static chamber (inner diameter 36 cm, height 100 cm) is specially made of transparent Plexiglas cylinder, but covered with outer tin foil to create dark environment. To homogeneously mix the inside gas when taking sample, we installed a fan on the top of chamber. Gas sampling was done 2, 4, 6, and 8 days after each N fertilization and floodwater drainage, and every ten days throughout the periods of water spinach growing. Gas sample collection was done between 7:00 and 9:00 a.m. We collected four gas samples using a 50 mL medical syringe 0, 15, 30 and 45 min after the static box was sealed for 15 min. At the same time, inside air temperature was recorded in each chamber. The N_2_O concentration was analyzed one week after gas sampling using a gas chromatograph (7890 A, Agilent Technology, Santa Clara, CA, USA) equipped with an electron capture detector, and the detection temperature was at 350 °C. The N_2_O flux and water spinach seasonal N_2_O emission were sequentially calculated using the cumulative summation method that detailed described in our previous work [37].

#### 4.3.3. The pH, EC, NH_4_^+^-N, NO_3_^−^-N, Organic and Total N Concentrations in Leachate

At the 60 cm depth of each soil column, one outlet was set up to collect the leachate sample. We sampled the leachate three times on 30 July, 11 August, and 29 August 2021, respectively. At each collection, we first recorded the volume of leachate from each soil column and then restored 100 mL water sample after filtration. The pH and EC of the leachate sample were measured by a pH and conductivity meter. Then, the NH_4_^+^-N, NO_3_^−^-N and total N concentrations were analyzed with a Continuous and Automatic Flow Injection Analyzer (SKALAR San^++^ System, Netherlands). The organic N concentration was calculated as the total N minus NH_4_^+^-N and NO_3_^−^-N.

### 4.4. Data Statistics

One way ANOVA and Duncan’s post hoc test were used to compare the differences between treatments at a significance level of *p* < 0.05, representing with different letters. Statistical analysis, as well as the mean and standard deviation (SD) of data (*n* = 3) were performed using SPSS 16.0 (SPSS Institute Inc., Chicago, IL, USA).

## 5. Conclusions

We conducted a soil column experiment to determine the after-effects of hydrochar inputs with 0.5 and 2.0 wt% on the production and N uptake of water spinach, and on the N leaching and N_2_O emission from vegetable soils with three N inputs. Hydrochar application at a high rate of 2.0 wt% could increase the biomass yield of water spinach receiving 20–40% reduced inorganic N fertilizer (120–160 kg/ha), which can be attributed to the higher total N recovery. Neither pH or EC was changed with the N reduction and hydrochar application. However, hydrochar addition disturbed the NH_4_^+^-N, NO_3_^−^-N, organic N, and total N concentrations in the leachate. In some cases, leaching N losses were increased by the hydrochar. Hydrochar application at both rates increased the cumulative N_2_O flux under 200 kg N/ha, but decreased it under 160 kg N/ha.

## Figures and Tables

**Figure 1 plants-11-03444-f001:**
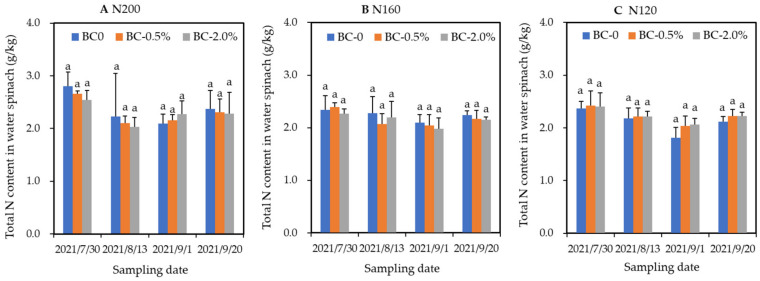
Responses of N contents of water spinach to hydrochar amendments under three N-inputs, i.e., 200 kg/ha (**A**), 160 kg/ha (**B**). and 120 kg/ha (**C**). Error bars represent the SD of the mean of three replicates. Identical letters above each column in the chart indicate the differences between each treatment with same N input were not statistically significant, according to Duncan’s post hoc test at level of *p* < 0.05.

**Figure 2 plants-11-03444-f002:**
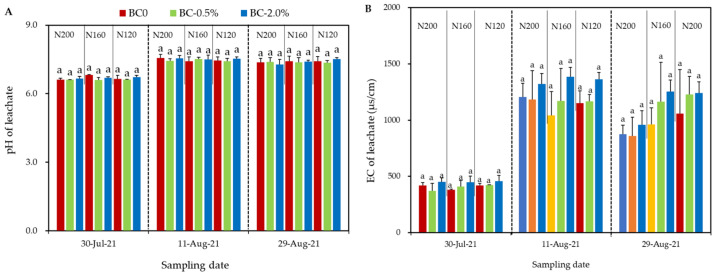
The pH (**A**) and EC (**B**) of leachate sampled at three observations on 30 July, 11 August, and 29 August 2021. Error bars represent the SD of the mean of three replicates. Same letters above column chart indicate the differences between each treatment with same N input were not statistically significant, according to Duncan’s post hoc test at level of *p* < 0.05.

**Figure 3 plants-11-03444-f003:**
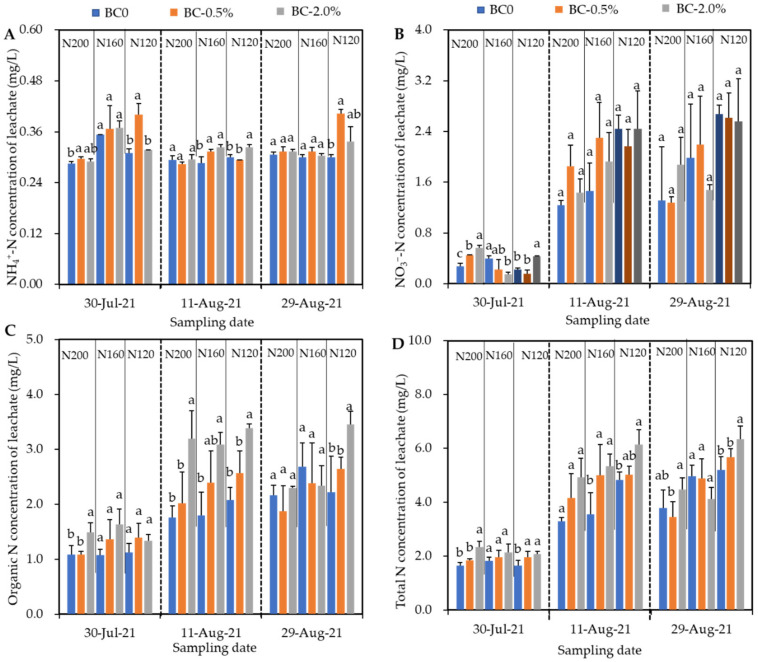
The NH_4_^+^-N (**A**), NO_3_^−^-N (**B**), organic (**C**) and total N (**D**) of leachate sampled at three observations on 30 July, 11 and 29 August 2021. Error bars represent the SD of the mean of three replicates. Different (same) letters above column chart indicate the differences between each treatment with same N input were (not) statistically significant, according to Duncan’s post hoc test at level of *p* < 0.05.

**Figure 4 plants-11-03444-f004:**
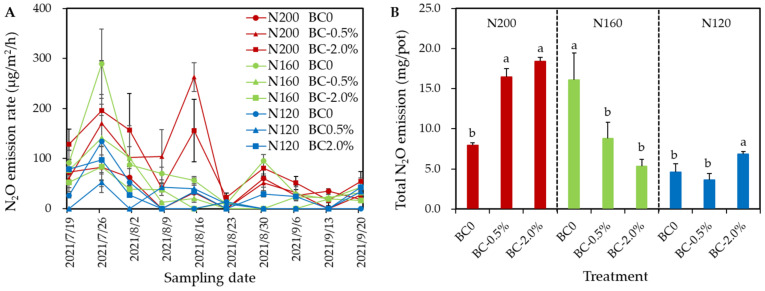
Effects of biochar on the N_2_O emission rate (**A**) and the total N_2_O emission flux (**B**) in a water-spinach-planted vegetable soil receiving three N inputs. Error bars represent the SD of the mean of three replicates. Different letters above column chart indicate the differences between each treatment with same N input were statistically significant, according to Duncan’s post hoc test at level of *p* < 0.05.

**Table 1 plants-11-03444-t001:** Responses of production (shoot biomass) of water spinach to hydrochar amendments under three N-inputs.

Treatment		Shoot Biomass of Water Spinach Plant at Different Sample Date (Fresh Weight, g/pot)				
		30 July 2021	13 August 2021	1 September 2021	20 September 2021	Total
N200	BC0	89.0 ± 36.2 a	110.1 ± 34.3 a	272.9 ± 36.8 a	196.5 ± 21.5 a	668.4 ± 58.4 a
	BC-0.5%	89.6 ± 29.4 a	129.7 ± 52.4 a	288.2 ± 40.2 a	201.0 ± 44.5 a	708.5 ± 48.5 a
	BC-2.0%	104.9 ± 27.7 a	139.8 ± 21.9 a	288.6 ± 48.6 a	205.9 ± 13.8 a	738.8 ± 18.6 a
N160	BC0	130.5 ± 47.3 a	96.8 ± 17.1 a	234.7 ± 15.1 a	189.0 ± 32.4 a	650.4 ± 53.7 b
	BC-0.5%	84.4 ± 36.1 a	106.1 ± 27.9 a	243.7 ± 12.7 a	206.5 ± 52.7 a	640.8 ± 62.9 b
	BC-2.0%	103.1 ± 21.3 a	136.7 ± 18.9 a	278.8 ± 39.4 a	207.3 ± 19.0 a	725.9 ± 56.7 a
N80	BC0	70.9 ± 16.7 b	108.4 ± 15.2 b	221.0 ± 13.9 a	215.5 ± 45.7 a	615.8 ± 52.4 b
	BC-0.5%	64.3 ± 21.3 b	139.2 ± 13.6 ab	191.6 ± 15.6 a	197.1 ± 34.6 a	592.3 ± 36.7 b
	BC-2.0%	119.9 ± 12.3 a	155.3 ± 16.7 a	206.6 ± 18.5 a	221.3 ± 18.7 a	703.0 ± 12.1 a

Note: Data were presented as mean ± SD (*n* = 3). Different letters in same column indicate the differences between each treatment with same N input were statistically significant, according to Duncan’s post hoc test at level of *p* < 0.05.

**Table 2 plants-11-03444-t002:** Effects of hydrochar amendment at two rates on N uptake capacity of water spinach under three N inputs.

Treatment		N Uptake Capacity of Water Spinach at Different Sample Time (mg/pot)				
		30 July 2021	13 August 2021	1 September 2021	20 September 2021	Total
N200	BC0	42 ± 8 a	44 ± 14 a	102 ± 11 a	83 ± 3 a	272 ± 7 a
	BC-0.5%	42 ± 11 a	50 ± 22 a	113 ± 27 a	84 ± 22 a	289 ± 76 a
	BC-2.0%	49 ± 18 a	51 ± 14 a	116 ± 12 a	85 ± 13 a	301 ± 47 a
N160	BC0	54 ± 20 a	39 ± 3 a	88 ± 6 a	76 ± 10 a	258 ± 18 a
	BC-0.5%	37 ± 17 a	40 ± 12 a	90 ± 12 a	81 ± 21 a	247 ± 44 a
	BC-2.0%	42 ± 10 a	54 ± 6 a	99 ± 20 a	80 ± 5 a	276 ± 27 a
N120	BC0	30 ± 6 b	43 ± 3 a	72 ± 7 a	83 ± 19 a	227 ± 22 b
	BC-0.5%	27 ± 7 b	56 ± 17 a	70 ± 7 a	78 ± 10 a	232 ± 4 b
	BC-2.0%	52 ± 8 a	62 ± 3 a	77 ± 3 a	89 ± 9 a	279 ± 10 a

Note: Data were presented as mean ± SD (*n* = 3). Different letters in same column indicate the differences between each treatment with same N input were statistically significant, according to Duncan’s post hoc test at level of *p* < 0.05.

**Table 3 plants-11-03444-t003:** Effects of hydrochar amending at two rates on NH_4_^+^-N, NO_3_^−^-N, organic and total N leaching losses from water-spinach-planted vegetable soil under three N inputs.

N Losses	Treatment		Sampling Date			Total
g/pot			21 July 2021	11 August 2021	29 August 2021	
NH_4_^+^-N	N200	BC0	0.34 ± 0.02 a	0.36 ± 0.02 a	0.37 ± 0.03 a	1.07 ± 0.07 a
		BC-0.5%	0.35 ± 0.02 a	0.33 ± 0.02 a	0.37 ± 0.04 a	1.06 ± 0.08 a
		BC-2.0%	0.37 ± 0.04 a	0.37 ± 0.04 a	0.39 ± 0.04 a	1.13 ± 0.13 a
	N160	BC0	0.42 ± 0.09 a	0.34 ± 0.04 b	0.35 ± 0.05 a	1.11 ± 0.17 a
		BC-0.5%	0.45 ± 0.04 a	0.39 ± 0.03 ab	0.39 ± 0.04 a	1.22 ± 0.11 a
		BC-2.0%	0.50 ± 0.03 a	0.44 ± 0.01 a	0.41 ± 0.02 a	1.34 ± 0.05 a
	N120	BC0	0.30 ± 0.01 b	0.29 ± 0.01 b	0.29 ± 0.02 b	0.87 ± 0.02 b
		BC-0.5%	0.47 ± 0.00 a	0.34 ± 0.01 a	0.47 ± 0.17 a	1.27 ± 0.16 a
		BC-2.0%	0.39 ± 0.10 ab	0.39 ± 0.07 a	0.41 ± 0.06 ab	1.19 ± 0.23 a
NO_3_^−^-N	N200	BC0	0.33 ± 0.02 c	1.50 ± 0.44 a	1.59 ± 0.18 a	3.42 ± 0.59 a
		BC-0.5%	0.53 ± 0.01 b	2.20 ± 0.38 a	1.54 ± 0.64 a	4.27 ± 0.97 a
		BC-2.0%	0.71 ± 0.08 a	1.83 ± 0.62 a	2.45 ± 1.37 a	4.99 ± 1.90 a
	N160	BC0	0.47 ± 0.18 a	1.71 ± 0.64 b	2.31 ± 0.80 a	4.49 ± 1.25 a
		BC-0.5%	0.28 ± 0.06 b	2.82 ± 0.62 a	2.69 ± 0.11 a	5.79 ± 0.71 a
		BC-2.0%	0.20 ± 0.04 b	2.59 ± 0.33 ab	1.99 ± 0.22 a	4.78 ± 0.32 a
	N120	BC0	0.21 ± 0.07 b	2.34 ± 0.23 a	2.56 ± 0.29 a	5.11 ± 0.48 a
		BC-0.5%	0.19 ± 0.01 b	2.53 ± 0.71 a	3.05 ± 0.79 a	5.76 ± 1.45 a
		BC-2.0%	0.52 ± 0.07 a	2.90 ± 0.45 a	3.11 ± 0.59 a	6.52 ± 0.72 a
Organic N	N200	BC0	1.31 ± 0.08 b	2.14 ± 0.76 b	2.63 ± 0.68 b	6.08 ± 1.27 b
		BC-0.5%	1.27 ± 0.10 b	2.40 ± 0.74 b	2.21 ± 0.16 b	5.89 ± 0.77 b
		BC-2.0%	1.87 ± 0.20 a	4.05 ± 0.91 a	2.85 ± 0.22 b	8.77 ± 0.82 a
	N160	BC0	1.27 ± 0.52 b	2.11 ± 0.67 b	3.11 ± 0.74 a	6.49 ± 0.44 b
		BC-0.5%	1.68 ± 0.42 b	2.93 ± 0.06 b	2.91 ± 0.23 a	7.52 ± 0.27 b
		BC-2.0%	2.20 ± 1.29 a	4.14 ± 0.19 a	3.15 ± 0.87 a	9.49 ± 0.94 a
	N120	BC0	1.07 ± 0.20 b	2.01 ± 0.46 b	2.14 ± 0.29 b	5.21 ± 0.54 c
		BC-0.5%	1.62 ± 0.12 a	2.98 ± 0.10 b	3.07 ± 0.30 b	7.68 ± 0.46 b
		BC-2.0%	1.62 ± 0.23 a	4.07 ± 0.50 a	4.16 ± 0.52 a	9.85 ± 1.07 a
Total N	N200	BC0	1.98 ± 0.10 b	4.00 ± 1.21 b	4.59 ± 0.88 ab	10.6 ± 1.90 a
		BC-0.5%	2.16 ± 0.08 b	4.94 ± 1.12 ab	4.12 ± 0.83 d	11.2 ± 1.75 a
		BC-2.0%	2.95 ± 0.32 a	6.25 ± 1.49 a	5.69 ± 1.20 a	14.9 ± 2.82 a
	N160	BC0	2.16 ± 0.50 b	4.16 ± 1.31 b	5.78 ± 0.25 a	12.1 ± 1.69 b
		BC-0.5%	2.41 ± 0.51 ab	6.14 ± 0.57 a	5.98 ± 0.14 a	14.5 ± 0.59 ab
		BC-2.0%	2.89 ± 0.34 a	7.17 ± 0.37 a	5.54 ± 0.66 a	15.6 ± 1.06 a
	N120	BC0	1.58 ± 0.15 b	4.63 ± 0.49 b	4.99 ± 0.32 b	11.2 ± 0.62 b
		BC-0.5%	2.27 ± 0.11 a	5.85 ± 0.67 ab	6.59 ± 0.56 a	14.7 ± 1.07 a
		BC-2.0%	2.52 ± 0.35 a	7.36 ± 0.75 a	7.67 ± 1.09 a	17.6 ± 1.85 a

Note: Data were presented as mean ± SD (*n* = 3). Different letters in same column indicate the differences between each treatment with same N input were statistically significant, according to Duncan’s post hoc test at level of *p* < 0.05.

## Data Availability

All data included in the main text.

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
