# Peer review of "After-Effects of Hydrochar Amendment on Water Spinach Production, N Leaching, and N2O Emission from a Vegetable Soil under Varying N-Inputs"

_plants, 2022, doi:10.3390/plants11243444_

Round 1

Reviewer 1 Report

 The study on hydrochar amendment effect on water spinach plants is interested, although there are some descriptions of hydrochar effect on water spinach in the scientific literature. The authors seems to try keep a consistency of the study results, but I have a problem with the figures descriptions. The description of the Fig.1 is a little suspicious – the authors wrote about variation in N content- when the letters located  above each  bar/column located in the figure are all the same, and the authors wrote in description of this figure that: ”different letters above column chart indicate the differences”.

I have also problem with the next figures:

-The Figure 2 – the authors in  the results description of the Fig. 2 wrote that “The leachate samples collected on July 30, 2021 were with relative lower pH and EC 116 values than that collected on August 11 and 29, 2021”, and the figure description contains statement: “ Different letters above column chart indicate the differences between each treatment with same N  input were statistically significant”, and all letters above the bars are the same. So,  therefore it is not clear if this equal N treatment included comparison of time of harvesting and hydrochar amendment, together.

-The Figure 3 – results descriptions also need more intelligibility.

So, the authors should correct their statistical calculations or descriptions of the figures, to be sure about correctness of presented conclusions.

Reviewer 2 Report

Biochar use in agriculture is one of very important issues in biochar studies. It is interesting and useful that the authors have investigated after-effects of hydrochar amendment on water spinach production, N leaching, and N2O emission from a vegetable soil under varying N-inputs. In totally, the MS was written sound. Hence, it is recommended to be published after minor revisions.

1.     P24-26, Change “but function varied impacts on N leaching and N2O emission, which are depending on inorganic N and hydrochar application rates” to “but impacts on N leaching and N2O emission varied, depending on inorganic N and hydrochar application rates”.;

2.     delete “Different letters above column chart indicate the differences between each treatment with same N input were statistically significant, according to Duncan’s post hoc test at level of p < 0.05.”, since no any different letters above column” in Figs 2&3. captions.
